

# Decadal comparison of a diminishing coral community: a study using demographics to advance inferences of community status

Margaret Miller[1], Dana E. Williams[1,2], Brittany E. Huntington[1,2,3], Gregory A. Piniak[4] and Mark J.A. Vermeij[5,6]

[1] Southeast Fisheries Science Center, NOAA-National Marine Fisheries Service, Miami, FL, United States
[2] Cooperative Institute for Marine and Atmospheric Studies, University of Miami, Miami, FL, United States
[3] Oregon Department of Fish and Wildlife, Newport, OR, United States
[4] National Centers for Coastal Ocean Science, NOAA-National Ocean Service, Silver Spring, MD, United States
[5] CARMABI Foundation, Willemstad, Curaçao
[6] Aquatic Microbiology/Institute of Biodiversity and Ecosystem Dynamics, University of Amsterdam, Amsterdam, NH, Netherlands

Corresponding author
Margaret Miller,
margaret.w.miller@noaa.gov

## ABSTRACT

The most common coral monitoring methods estimate coral abundance as percent cover, either via *in situ* observations or derived from images. In recent years, growing interest and effort has focused on colony-based (demographic) data to assess the status of coral populations and communities. In this study, we relied on two separate data sets (photo-derived percent cover estimates, 2002–12, and opportunistic *in situ* demographic sampling, 2004 and 2012) to more fully infer decadal changes in coral communities at a small, uninhabited Caribbean island. Photo-derived percent cover documented drastic declines in coral abundance including disproportionate declines in *Orbicella* spp. While overall *in situ* estimates of total coral density were not different between years, densities of several rarer taxa were. *Meandrina meandrites* and *Stephanocoenia intersepta* increased while *Leptoseris cucullata* decreased significantly, changes that were not discernable from the photo-derived cover estimates. Demographic data also showed significant shifts to larger colony sizes (both increased mean colony sizes and increased negative skewness of size frequency distributions, but similar maximum colony sizes) for most taxa likely indicating reduced recruitment. *Orbicella* spp. differed from this general pattern, significantly shifting to smaller colony sizes due to partial mortality. Both approaches detected significant decadal changes in coral community structure at Navassa, though the demographic sampling provided better resolution of more subtle, taxon-specific changes.

# INTRODUCTION

Coral cover and community composition have been established as the standard metrics for reef monitoring programs. Visual census techniques have been around for decades (*Stoddart & Johannes, 1978*), and gained early recognition for their utility in understanding

population dynamics (*Hughes, 1984*) and disturbance/recovery dynamics (e.g., *Done, 1985*). These techniques provide a valuable, albeit general, overview of the status of reef communities, but offer little insight on the processes that drive the observed patterns. Coral cover also carries relatively low signal:noise ratio in depauperate reef areas (e.g., <10 % or even 1–2% cover) as is characteristic of many modern Atlantic/Caribbean reefs (*Jackson et al., 2014*), making change detection difficult without substantive (and perhaps unrealistic) increase in sample size (*Molloy et al., 2013*). As a result, there has been increasing interest (and implementation) in expanding from simple percent cover to more refined, process-based measures.

Coral demographic parameters have only recently been included in comprehensive coral monitoring programs but may be a valuable supplement to percent cover data, as a demographic approach offers species-specific and mechanistic insights into observed changes in percent cover. Regional programs in south Florida (e.g., *Smith et al., 2011*; and Florida Reef Resilience Program, frrp.org/coral-monitoring) and internationally (e.g., Atlantic and Gulf Rapid Reef Assessment: *Lang & Ginsburg, 2006*; IUCN: *Obura & Grimsditch, 2009*) began implementing colony based size and density measures in the early–mid 2000s. There is a wealth of ecological theory to support analysis and interpretation of coral colony-based demographic data (*Bak & Meesters, 1998*; *Vermeij & Bak, 2002*) whereby changes in the size structure of a population is used to infer underlying ecological processes. Meanwhile, the collection of such data over large programs and time frames carries some potential challenges such as inter-observer variation in detection (especially of small corals), uncertainty in species identification (especially of small colonies), and consistently delineating colony boundaries in populations with large amounts of partial mortality.

In this study, we analyze coral information derived from both percent cover data and from demographic data collected at haphazardly selected reef sites to examine temporal change in coral status at Navassa, a small, uninhabited Caribbean island. Reefs in this area have suffered severe disturbances, including hurricane impacts, severe disease outbreaks, and mass bleaching over the past decade resulting in a drastic loss of live coral cover (*Miller et al., 2008*). We present coral percent cover data using standard photo-quadrat techniques (collected in 2002, 2004, 2006, 2009, 2012) in parallel with a separate coral demographic data set collected in 2004 and 2012. Our purpose is not to compare between these sampling methods per se, as choice of method is influenced not only by practical considerations (level of effort, etc.) but also the type of information desired and applicability within a habitat type. Rather, we examine these data sets (the only *in situ* data available from reefs of this uninhabited island) in parallel to address two questions: (1) Are observed changes in coral community status consistent between these two data sets (i.e., cover versus colony density and size structure); and (2) Does the demographic data provide insights on processes underlying observed changes that are not evident from the cover data alone?

## METHODS

The small oceanic island of Navassa (18.40° N, −75.01° W) is a component of the United States National Wildlife Refuge system located approximately 55 km off the southwest

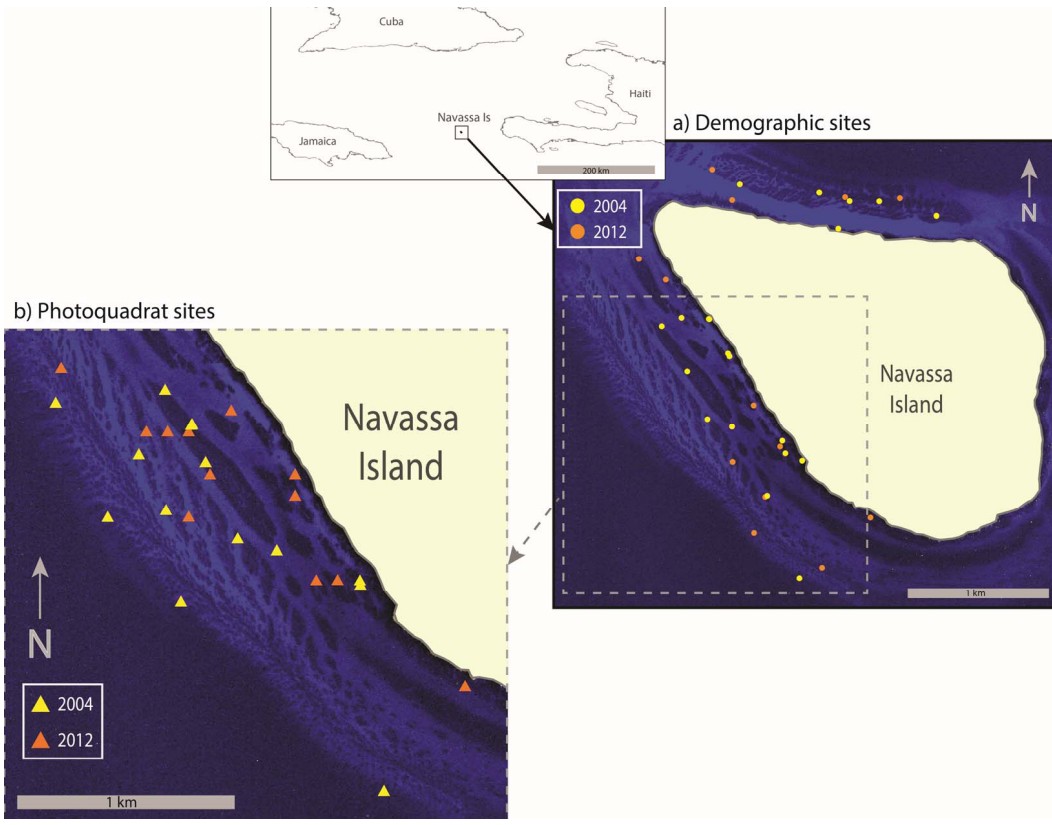

**Figure 1** **Sample Location.** Map showing location of Navassa Island in the greater Antilles. Insets show the specific location of demographic sites (A) and photoquadrat sites (B) that were sampled in 2004 and 2012. Photoquadrat sites sampled in the other years were similarly dispersed among reef habitats of the southwest coast (i.e., 2004 and 2012 are shown as representative examples). Site locations are superimposed on satellite imagery showing reef distribution along the Navassa shelf (IKONOS image provided by DigitalGlobe).

tip of Haiti. Though uninhabited, it is frequently visited by Haitian subsistence fishers. Reef development mostly occurs on the narrow shelf along the leeward southwest coast of the somewhat triangular island, whereas low-relief reef communities are found along the exposed north coast (Fig. 1). Benthic habitats along the windward east coast mainly consist of rubble bottom. Due to its remoteness, there has been no structured monitoring program at Navassa. Instead, episodic, opportunistic cruises have gathered both demographic and photoquadrat/cover data. The opportunistic cruises were not intentionally designed to provide a rigorous comparison across methodologies. Thus, here we leverage the only available data for a remote Caribbean reef system starting from a relatively unimpacted baseline (*Miller & Gerstner, 2002*) by laying out parallel observations derived from two separate coral sampling schemes, collected within coherent time and habitat strata, to examine the observed differences to determine what meaningful and complementary inferences can be discerned.

## Photoquadrat data

A set of sites along the southwest shelf of Navassa (depth 18–34 m), was sampled with haphazardly-placed photoquadrats in 2002, 2004, 2006, 2009, and 2012. In 2002 and 2004 the sites were haphazardly selected (but always targeting reef habitats along the southwest coast) by necessity, as no habitat maps were available. Within the logistic constraints of working from a single large ship with multiple cruise objectives, effort was made to disperse these sites throughout the southwest shelf reef habitats. Later, the development of habitat maps from efforts on the 2004 cruise (*Miller, Halley & Gleason, 2008*) enabled the stratified-random selection of sampling sites in 2006, 2009, and 2012, similarly restricted to patch- and slope-reef habitats along the southwest coast in order to be comparable to sets of sites sampled in earlier years. Although the means of site selection varied between sample years, a relatively high sampling intensity (hence a representative sample given the restriction of habitat strata) was accomplished in each year (e.g., >2 sites km$^{-2}$ shelf area in 2004, >4 sites km$^{-2}$ shelf area in the other years, e.g., Fig. 1B).

Photoquadrats were placed by taking five fin kicks in a haphazard direction from the prior quadrat and then tossing the quadrat forward while the diver's eyes were closed, yielding approximately 6–8 m distance between quadrats. Each photoquadrat was comprised of a 1 m$^2$ image and a close-up of each quadrant of the quadrat (to provide a clearer image for reference to aid in coral identification), for a total of five images per quadrat. Benthic community structure was analyzed by overlaying each 1 m$^2$ picture with 100 (2002–4) or 50 (2006–12) random point counts (reduced over time because large numbers of points per frame do not contribute to improved power; *Aronson et al., 1994*; *Houk & Van Woesik, 2006*) using CPCe software (*Kohler & Gill, 2006*). Corals were identified to species when possible, genus if not. Eight to ten quadrats were analyzed per site.

Variation over time in percent live coral cover was analyzed for the community as a whole (all species summed) as well as for the most common individual taxa (*Orbicella* spp. (primarily *O. faveolata*), *Agaricia* spp. (primarily *A. agaricites*), and branching *Porites*) via separate one-way ANOVAs on ranks followed by Dunn's post-hoc tests for differences among individual years. To make parallel, taxa-specific comparisons in percent cover as for the density data (described below), we additionally performed separate Mann–Whitney rank sum tests to test for differences in cover for each of three rarer taxa (*Siderastrea siderea*, *Meandrina meandrites*, and *Stephanocoenia intersepta*) between 2004 and 2012.

## Coral demographic data

In 2004 and 2012, coral colonies were surveyed at a separate set of haphazardly selected reef sites (Table 1; including a subset of the stratified-random photoquadrat sites in 2012), with effort to disperse these sites among the patch reef and slope reef habitats of the southwest and north coasts (Fig. 1). This demographic sampling was accomplished via belt transects (15 m × 0.5 m (2004) or 10 m × 1 m (2012)), with short dive times due to deep depths sometimes dictating a smaller sample area. The actual area sampled was recorded for each transect and used to standardize colony density (as # colonies m$^{-2}$). Within each belt transect, every colony (defined as all tissue sharing a single skeletal unit, even if multiple live tissue isolates were divided by areas of dead skeleton) was identified to species, and

 

**Table 1  Summary effort and density.** In-water effort for 2004 and 2012 demographic ('Demo') and photoquadrat (PQ) sampling including number of sites, transects or photoquadrats and person-dives. Area (m²) indicates the cumulative area of transects (demo) or quadrats sampled. The total number of colonies sampled and the colony density (# colonies/m²) are also given under the demographic section.

| Year | Location | PQ # sites | PQ area | PQ person-dives | Demo # sites | Demo # transects | Demo person-dives | Demo area | # col | Overall density (SE) |
|------|----------|-----------|---------|-----------------|--------------|------------------|-------------------|-----------|-------|----------------------|
| 2004 | SW | 14 | 149 | 16 | 13 | 17 | 13 | 111.5 | 1,227 | 11.48 (1.06) |
|  | N[a] |  |  |  | 6 | 7 | 6 | 50.5 | 351 | 7.07 (0.56) |
| 2012 | SW | 12 | 125 | 13 | 8 | 18 | 20 | 166.5 | 1,711 | 10.17 (0.80) |
|  | N[a] |  |  |  | 4 | 6 | 5 | 51 | 316 | 6.18 (0.46) |

**Notes.**

[a] Transects from the north coast are included only in the size frequency analyses (Table 2 and Fig. 4).

size was recorded for most colonies (*Porites astreoides* was tallied for density but not sized in 2012 due to dive time constraints). In 2004, a clear acetate grid was overlaid the colony and used to estimate projected live colony area directly. In 2012, the dimensions of each colony (maximum diameter and the diameter perpendicular to the maximum) and a visual estimate of its projected % live area were recorded *in situ* as was deemed more consistent to apply among multiple observers and to conform to regionally established protocols (AGRRA; http://www.agrra.org/method/methodhome.html). To compare colony areas with those measured in 2004, a circular area (2-dimensional, projected) was estimated with a diameter that was the average of the two diameters measured, and adjusted for the estimated % live area of the colony (adjusted circular area). While the adjusted circular area and acetate grid are different means to estimate area, all field methods represent approximations and these are both reasonable, comparable methods. Other efforts specifically aimed and comparing different geometric approximations for coral colony size/area have shown negligible differences (e.g., between using a circular versus elliptical approximation for 2d colony area; *Van Woesik et al., 2011*). Colonies of less than 2 cm diameter (3.14 cm² area) were excluded from subsequent analyses to account for potential observer bias in the detection of small colonies and inherent difference in detectability between years or transects due to variable cover of the macroalga, *Lobophora variegata*. Identification uncertainties for the smaller juveniles also dictated pooling of certain taxa (mostly to genus, though *Porites* was delineated into branching and mounding morphologies; see list in Fig. 3).

For both coral density and multivariate community structure analyses, only transects located along the southwest (leeward) coast of the island between 18 and 34 m depth were included to standardize the sampled habitat in these analyses where including replicates in different habitats would increase variance and decrease power. A smaller sampling effort did occur along the north coast in each year (Table 1), but the reef habitats and benthic assemblage found here are substantively distinct (*Miller, Halley & Gleason, 2008*) relative to the more-developed southwest reefs. However, since the colonies along the north and southwest coasts clearly do not represent distinct populations (being separated by <1 km distance) and sampling effort was similar between years (Table 1), all colonies available (including the north coast) were pooled for the within-taxa size frequency analyses (described below) where pooling habitats helped boost sample size.

Coral density was tallied for each transect ($n = 17$ for 2004, 18 for 2012) along the southwest coast for 15 taxa in each year to generate mean abundance estimates of coral species found in Navassa's high relief reef habitat. Univariate Mann–Whitney rank sum tests (or $t$-tests) were used to test for univariate differences in density between years for each taxon and for total coral density using transects as replicates. To characterize potential differences in coral assemblage structure between 2004 and 2012, we calculated Bray–Curtis similarities on species density estimates among all transects after square-root transforming the data to reduce the influence of highly dominant taxa (*Clarke & Warwick, 2001*). A non-metric Multi-Dimensional Scaling (nMDS) plot was created to visualize differences in coral composition of transects between years while Analysis of Similarity (ANOSIM) was used to determine significance of these differences. Similarity Percentage (SIMPER) analysis was applied to identify which taxa were most influential in determining significant difference between years (PRIMERe v.6.0).

In analyzing potential differences in size structure of coral populations, we focussed on taxa with $n > 30$ colonies sized in each year. Colony areas were ln-transformed (*Vermeij & Bak, 2002*) and histograms constructed for each sampled year (2004 and 2012). Descriptive statistics were calculated and the distribution of colonies among size classes was compared between years via Kolmogorov–Smirnov tests for each taxa. Taxa not consistently measured in 2012 (due to dive time constraints (i.e., *Porites astreoides*)) are not included in analyses of size frequency.

Lastly, the individual colony areas (measured taxa only) for each transect were summed by taxa and divided by the transect area. This yields a demographically-derived estimate of coral cover (as coral area) to provide an integration of the (potentially contrasting) density and size differences among taxa. This allows visualization of contrasts in assemblage composition according to different 'currencies' (density versus area occupied) within the same demographic data set.

This research was conducted under Navassa National Wildlife Refuge Special Use Permits #41529-2002-10, #41529-2004-12 and #41529-2006-03, #41529-2009-01, and # 41529-2012-001 from the US Fish and Wildlife Service.

## RESULTS

### Photoquadrat data

The average photo-derived coral cover along the southwest coast of Navassa declined from 34% in 2002 to 9.6% in 2012. 2002 and 2004 do not differ significantly from each other, but they both are significantly higher than the subsequent three survey years (Fig. 2, Dunn's post-hoc pairwise comparisons). *Orbicella* spp. (predominantly *O. faveolata*) constituted about three-quarters (0.76) of total coral cover in 2002, but only one fifth (0.20) of coral cover in 2012 with the steepest (and statistically significant) decline between 2002 and 2004, prior to the demographic sampling (Fig. 2). In contrast, the other two taxa with the highest cover showed more gradual declines and retained similar proportional representation of total cover over the same time frame (0.20–0.23 for *Agaricia* spp. and 0.19–0.23 for branching *Porites*) (Fig. 2).

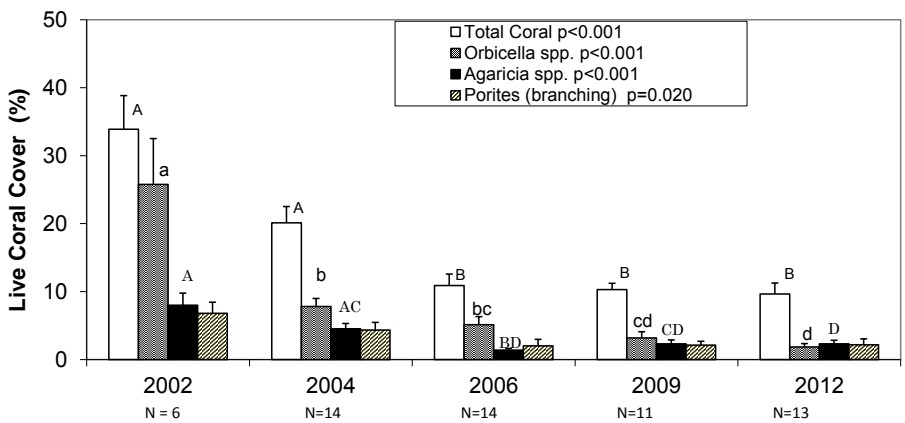

**Figure 2 Photo-derived percent cover.** Percent cover (+1SE) for total scleractinian corals and for the three most abundant coral taxa in Navassa photoquadrats over time. The three most abundant taxa are *Orbicella* spp. (predominantly *O. faveolata*), *Agaricia* spp. (dominated by *A. agaricites*), and branching *Porites* (*P. porites*, *P. furcata*, and *P. divaricata*). *P*-values in legend from separate one-way ANOVA on ranks for each taxa across time. *N* (number of sites) is given for each year under the axis. Similar letters over each set of bars indicate no statistical difference in post-hoc comparisons for a given taxa across time. Note that *Orbicella* spp. cover had declined most before 2004, whereas the other two taxa (and total coral cover) continued declining through 2006.

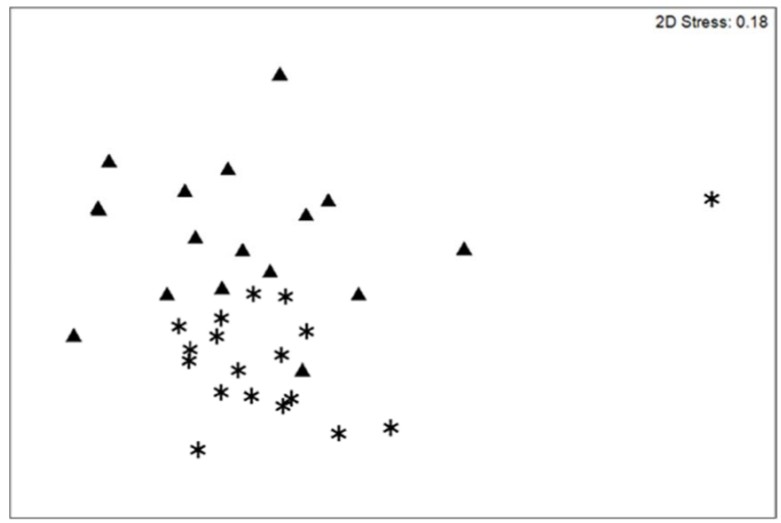

**Figure 3 Community change.** nMDS scaling plot for coral demographic transects sampled on the southwest coast of Navassa in 2004 (triangles) and 2012 (asterisks). Although the somewhat high stress level indicates that this 2-dimensional rendering is not a perfect representation of the similarity among samples, ANOSIM indicates significant change in coral community structure (based on square-root transformed colony density) between the two years (Global $R = 0.308$; significance level of 0.1%).

## Coral demographic data

In contrast to total photo-derived coral cover, total coral density along the southwest coast did not differ between 2004 and 2012 (Table 1, *t*-test $p = 0.33$). However, significant differences in species composition did occur (Global $R = 0.308$; significance level of 0.1%; Fig. 3) with *S. siderea*, *A. agaracites*, branching *Porites*, *Leptoseris cucullata*, and
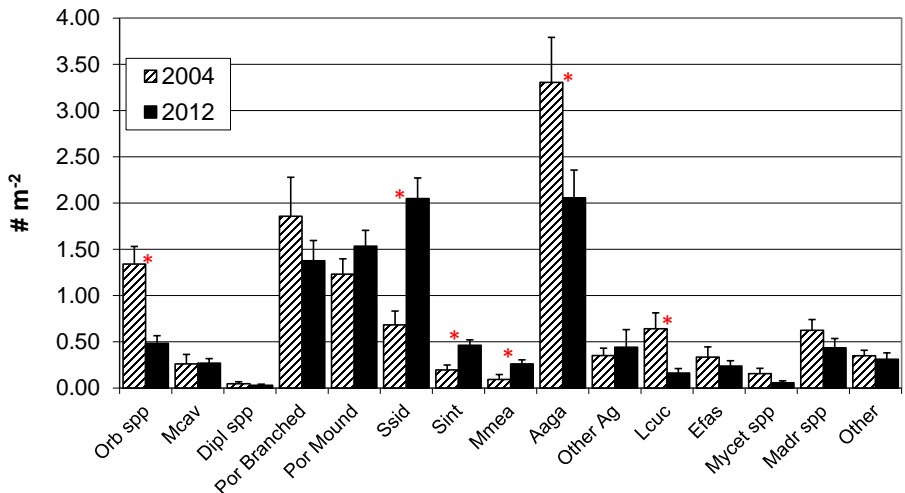

**Figure 4** **Colony density by taxa.** Colony density for coral taxa (mean + 1SE; *n* = 17, transects in 2004 and 18 in 2012) in the demographic sampling of sites along the southwest coast of Navassa. *indicates significant difference between the two years as determined by Mann–Whitney rank sum test ($p < 0.05$). Orb spp, *Orbicella* spp. (predominantly *O. faveolata*); Mcav, *Montastraea cavernosa*; Dipl spp, *Diploria* (includes recently reclassified *Pseudodiploria strigosa*, *Budd et al., 2012*) species; Por branched, branched *Porites* spp (*P. porites, P. furcata, P. divaricata*); Por Mound, mounding *Porites* species (predominantly *P. astreoides*); Ssid, *Siderastrea siderea*; Sint, *Stephanocoenia intersepta*; Mmea, *Meandrina meandrites*; Aaga, *Agaricia agaricites*; Other Ag, *Agaricia* species other than *A. agaricites*; Lcuc, *Leptoseris cucullata*; Efas, *Eusmilia fastigiata*; Mycet spp, *Mycetophyillia* species; Madr spp, *Madracis* species; Other, other scleractinians.

*Orbicella* spp. being the taxa contributing foremost to the dissimilarity between the two years, cumulatively contributing 45% of the total dissimilarity (SIMPER analysis). When analyzed separately (univariate Mann–Whitney rank sums tests), *A. agaricites, L. cucullata,* and *Orbicella* spp. all showed significant univariate decreases in density whereas branching *Porites* did not show any significant difference between 2004 and 2012 (Fig. 4). Coral species exhibiting significantly higher colony density in 2012 included one common species, *S. siderea*, and two rarer species, *Stephanocoenia intersepta* (formerly *S. michelini*) and *Meandrina meandrites* (Fig. 4). In comparison, the photo-derived coral cover similarly captured a significant increase between 2004 and 2012 in *S. siderea* (from 0.41 to 0.97% cover; $U = 57$, $p = 0.039$), but no significant change was detected for *M. meandrites* (0.10–0.20%; $U = 100$, $p = 0.864$) nor *S. intersepta* which was not identified in any of the photoquadrats in 2012, though it was detected at low abundance (a mean of .013%) in 2004.

Robust coral size frequency distribution comparisons ($n \geq 100$ colonies) were obtained for four taxa and for another four taxa based on smaller sample sizes ($n = 30$–99) (Table 2; three additional taxa were recorded but lacked sufficient sample size for comparisons between years). All four of the taxa with larger sample sizes show significant changes in size frequency distributions between 2004 and 2012 (Figs. 5A–5D and Table 2). *Orbicella* spp. populations showed smaller mean colony size (985 cm² in 2004 to 347 cm² in 2012), while branched *Porites* spp., *S. siderea,* and *A. agaracites* had larger mean colony sizes. Of the four taxa with lower sample size, only *L. cucullata* showed a significant difference (Fig. 5E
**Table 2  Size frequency data.** Coral colony size frequency summary statistics from Navassa demographic sampling in 2004 and 2012, based on ln (colony area (cm$^2$)). *P*-values are given for Kolmogorov–Smirnov tests comparing distributions between the two years. Three sections indicate groups of taxa with progressively lesser *N*'s. Histograms for taxa that are significantly different are shown in Fig. 5. Taxa as in Fig. 4.

| | | 2004 | | | | | | 2012 | | | | | | | |
|---|---|---|---|---|---|---|---|---|---|---|---|---|---|---|---|
| | | Count | Mean | Median | Var | S.D. | Skewness | Count | Mean | Median | Var | S.D. | Skewness | p | KS |
| I | Aaga | 438 | 3.89 | 3.74 | 2.24 | 1.50 | 0.24 | 365 | 4.28 | 4.46 | 1.48 | 1.22 | −0.53 | <0.001 | 3.51 |
| | Orb spp | 183 | 5.82 | 5.99 | 2.64 | 1.62 | −0.42 | 100 | 5.08 | 5.45 | 2.22 | 1.49 | −0.77 | <0.001 | 2.04 |
| | Por Branched | 212 | 2.73 | 2.20 | 2.09 | 1.45 | 1.36 | 199 | 4.32 | 4.64 | 1.94 | 1.39 | −0.45 | <0.001 | 5.30 |
| | Ssid | 154 | 2.85 | 2.48 | 1.63 | 1.28 | 1.18 | 422 | 3.12 | 2.98 | 1.59 | 1.26 | 0.47 | <0.001 | 2.32 |
| II | Efas | 36 | 3.44 | 2.94 | 2.96 | 1.72 | 0.61 | 45 | 3.70 | 3.50 | 2.27 | 1.51 | 0.87 | 0.31 | 0.97 |
| | Lcuc | 87 | 3.05 | 3.04 | 1.18 | 1.09 | 0.27 | 30 | 3.70 | 4.14 | 1.95 | 1.40 | −0.37 | 0.00 | 1.76 |
| | Mcav | 31 | 3.63 | 3.30 | 3.36 | 1.83 | 0.78 | 46 | 3.59 | 3.42 | 2.21 | 1.49 | 0.30 | 0.81 | 0.64 |
| | Sint | 41 | 3.40 | 3.30 | 1.51 | 1.23 | 0.53 | 96 | 3.53 | 3.50 | 1.54 | 1.24 | −0.03 | 0.19 | 0.83 |
| III | Dipl spp[a] | 5 | 6.95 | 7.90 | 2.60 | 1.61 | 0.46 | 5 | 6.49 | 6.63 | 2.21 | 1.49 | 0.07 | n/a | n/a |
| | Mmea | 12 | 4.36 | 4.43 | 1.45 | 1.21 | −0.17 | 57 | 4.63 | 4.73 | 2.17 | 1.47 | −0.43 | n/a | n/a |
| | Mycet spp | 18 | 3.54 | 3.47 | 1.58 | 1.26 | 0.46 | 11 | 3.37 | 3.65 | 1.67 | 1.29 | 0.07 | n/a | n/a |

**Notes.**

[a] Includes recently reclassified *Pseudodiploria strigosa* (*Budd et al., 2012*).

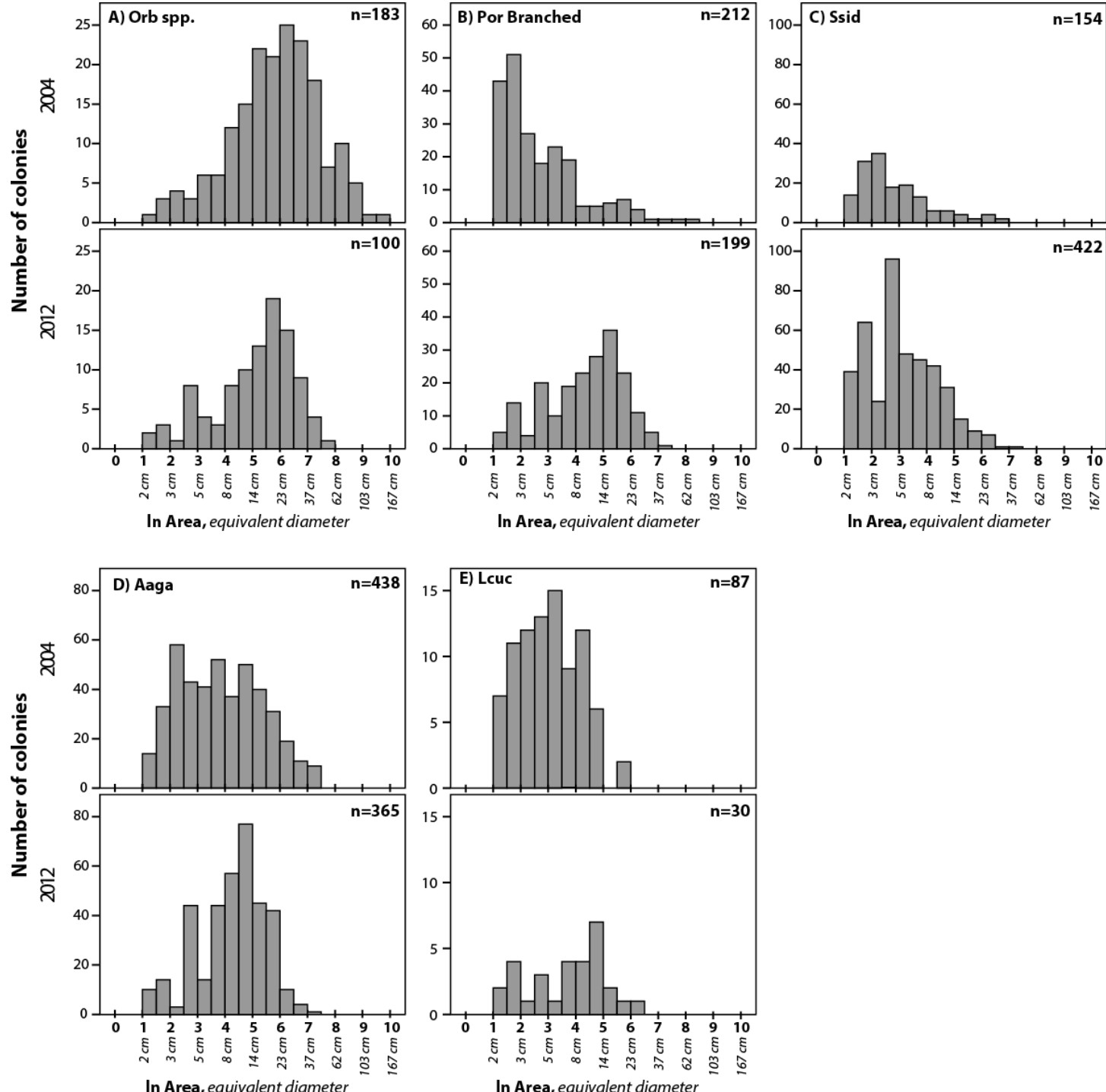

**Figure 5** **Size frequency distributions for five taxa that showed significant difference between 2004 and 2012.** Size bins are expressed as ln (colony area in cm²). Corresponding colony diameters (for calculated circular area) are shown below the *x*-axis for scale. Summary statistics given in Table 2. Taxa as in Fig. 4.

and Table 2), in this case shifting to a larger mean colony size. Skewness shifted from positive to negative for four of the taxa (*A. agaricites*, branched *Porites*, *L. cucullata*, and *S. intersepta*) indicating increased under-representation of small (or over-representation of large) colonies in these populations (Table 2; *Vermeij et al., 2007*).

Pooling the area occupied by measured corals, juxtaposed with their cumulative density (Fig. 6) aids in integrating the contrasting patterns of decreased cover, stable density, and contrasting size changes among taxa. The calculated area occupied by these corals in 2012 is less than half of that in 2004, despite only small (insignificant) change in colony density. Additionally, individual taxa show contrasting patterns between area and density units, due to the influence of different size structure. While *Orbicella* spp. showed small losses (a factor of 0.36) in density between 2004 and 2012, its losses in calculated area were extreme (less than one tenth remained due to loss of large colonies). In contrast, *S. siderea* showed much greater increases in density (tripled) than in area occupied (due to increased abundance constituted by small colonies occupying little area). Meanwhile, some taxa (e.g., *A. agaricites*) showed approximately proportional changes in both units (Fig. 6).

## DISCUSSION

The overall decline in coral cover and lack of resilience displayed among Caribbean reefs over the past decades is well described in the literature. A large meta-analysis covering sites throughout the Caribbean from 1970 to 2012 (*Jackson et al., 2014*) provides context for the changes described here in Navassa reefs in the later portion of this interval. This meta-analysis indicates the mean corrected coral cover for deeper reefs (5–20 m depth, 88 locations) over three time periods declined from 32.6% (1970–1983) to 19.4% (1984–1998) to 16.5 (1999–2012) (*Jackson et al., 2014*, Part 1, Table 3, p. 67). Though most of our Navassa sampling sites in all years were deeper than 20 m, the initial coral cover documented at Navassa's southwest reefs in this study was 34% in 2002, corresponding with the Caribbean-wide average several decades earlier. Subsequently, a 20% absolute decline in Navassa coral cover occurred over a period of just four years, whereas the Caribbean-wide mean decline of only 16% absolute took two decades. While coral cover at Navassa does appear to have remained robust for a longer duration in the absence of local development and human habitation, acute disturbance events of global stressors such as thermally-induced coral bleaching (*Miller, Piniak & Williams, 2011*) and coral disease (*Miller & Williams, 2006*) have coincided at Navassa with at least as great a magnitude of decline at a much more rapid pace than the regional average.

We acknowledge several caveats to the data presented here when interpreting our findings. The sites sampled in most cases were haphazardly chosen. However, all sites sampled in both data sets were constrained by habitat type for each temporal comparison (e.g., to deep patch and bank reefs on the southwest coasts for coral density). The total shelf area at Navassa is small and the relative density of sampled sites was adequate to detect differences between years in both cover and demographic parameters. The demographic data were collected opportunistically to supplement % cover monitoring data, not as planned repeat monitoring or a methods comparison. Furthermore, our sampling in
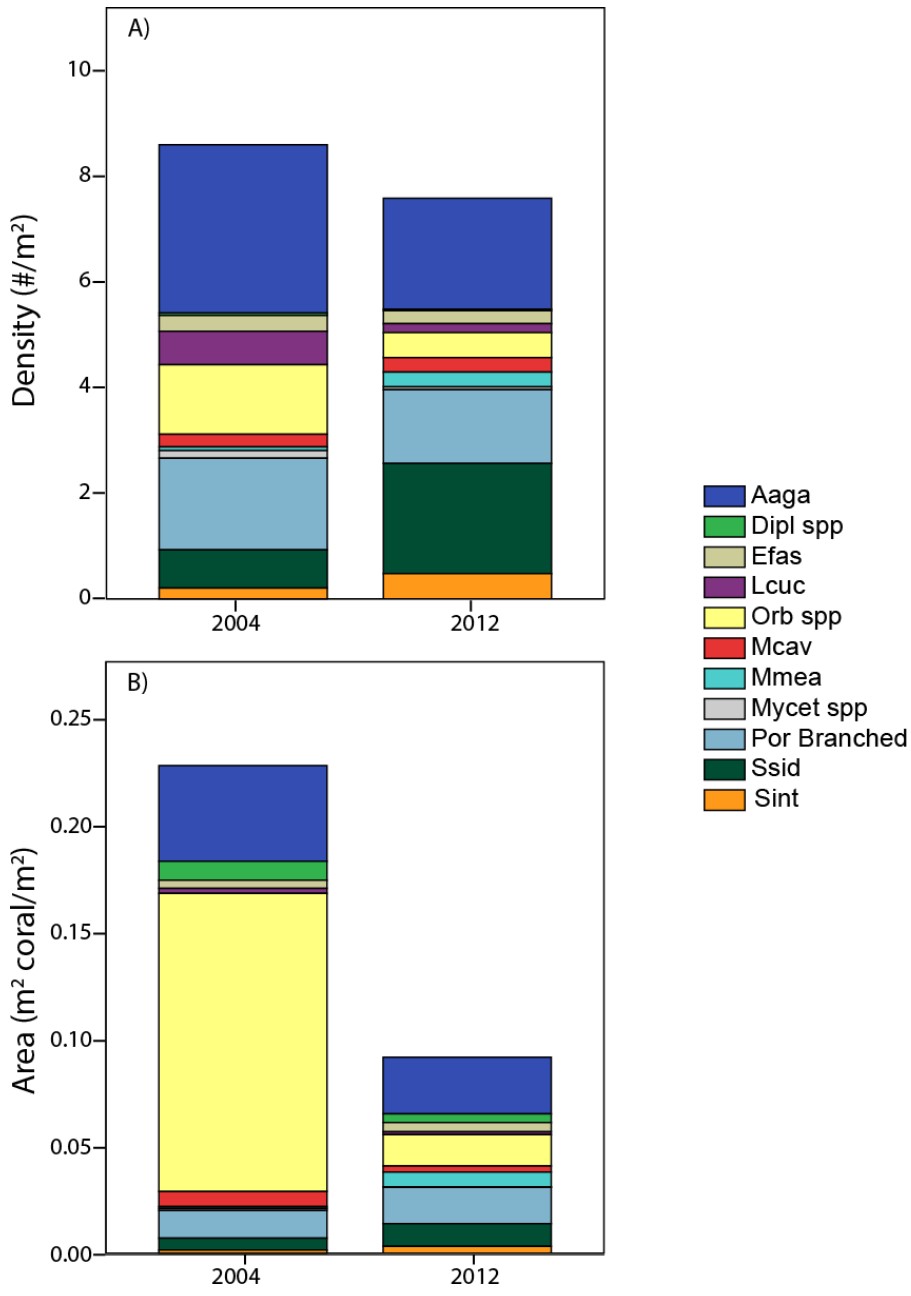

**Figure 6 Area occupied by taxa.** A comparison of coral composition (sized taxa only) for southwest sites based on demographic data expressed as density (A) and as area back-calculated by summing the area occupied by each of these sized colonies (B). Taxa abbreviations as in Fig. 4.

both years was constrained by a limited number of dives per cruise to survey the coral community, not shaped by an objective power analysis nor optimal spatial allocation of samples (e.g., *Smith et al., 2011*). The sampling effort (i.e., person dives and area covered) was fairly similar between both data sets in 2004, but somewhat higher for the demographic data in 2012 (Table 1). Given this non-homogenous field sampling, we took a conservative approach to our analyses. Our analytical approach has been to filter the demographic

data in conservative ways to avoid potential bias in the primary comparison between years i.e., excluding (e.g., small sizes) or pooling (e.g., nominal spp identifications into higher taxa groups) data whenever there was uncertainty in comparability. Given this sub-optimal data set, interesting and significant differences between years are evident from the demographic data (Fig. 4 and Table 2) that are not resolvable from the more traditional and more frequent coral cover data.

Significant loss in coral cover occurred while total coral density remained unchanged, as substantially lower colony abundance of some taxa (mainly *Orbicella* spp) was offset by significantly higher density of other, typically smaller taxa. These taxa with higher density include both the common *S. siderea* (for which increase was also apparent from the more frequent photo-derived cover data), as well as two rare species for which changes in photo-derived cover were not discernable (*M. meandrites* and *S. intercepta*). High recruitment rates have similarly been reported for these three species in other studies (e.g., *Huntington & Lirman, 2012*; *Vermeij et al., 2011*). Historic studies of juvenile coral assemblages in Curaçao indicate *M. meandrites* ranking 4th and *S. intersepta* 7th in terms of relative abundance within the total juvenile population (*Bak & Engel, 1979*). For *M. meandrites*, our 2004 mean density of 0.09 colonies m$^2$ is congruent with that reported by *Pinzon & Weil (2011)* measured in southwest Puerto Rico in 2002–3, as is a negative skewness of its size distribution. However, this species showed a decrease in juvenile density in Curaçao between 1979 and 2005 (*Vermeij et al., 2011*), and we are unaware of other published reports of significant increasing abundance trends for this species as observed at Navassa (Fig. 4). Interestingly, *L. cucullata*, a significant 'loser' in colony abundance in the current study, has showed a similar drastic decrease in juvenile density in both Curaçao (as *Helioseris cucullata*; (*Vermeij et al., 2011*)) and Jamaica (*Hughes & Tanner, 2000*).

The size frequency data showed that some species shifted significantly toward larger colonies, and others toward smaller size. The loss of very large *Orbicella* spp. colonies is most likely attributable to substantial mortality associated with disease and bleaching events between 2004 and 2006 (*Miller, Piniak & Williams, 2011*; *Miller & Williams, 2006*). The most drastic difference in size distribution among the taxa examined was a strong shift to larger colonies in branching *Porites*. The reason for this remains unclear, but a reduction in recruitment combined with growth of colonies through time (from a modal diameter of 3 cm to a modal diameter of 14 cm in eight years, Fig. 5B) seems plausible, given no significant change in density (Fig. 4). Alternatively, it is possible that a substantial shift in species representation within this morphological group may have occurred (e.g., more *P. porites* and less of the small *P. furcata* and/or *P. divaricata*). We do not think this explanation likely due to substantial representation by both small and larger morphs in both sampling years, and recent genetic evidence has failed to support these three as distinct species (*Prada et al., 2014*).

We considered the sampling efficiency of collecting both percent cover and demographic data in deep water reefs where bottom time is limiting. The present data sets were collected with roughly comparable levels of in-water effort per unit sample (at the depths and dive times available for these deep reefs, a single diver was able to complete either one

demography transect or a set of photoquads, each method surveying $\sim 10$ m$^2$ of the seafloor, Table 1), though there are clear tradeoffs in both the amount of post-processing effort required and aspects of statistical power related to ten replicate photoquads versus a single transect that yields data on a large but variable number of coral colonies. The consequences of these replication differences will depend on the types of analyses attempted. Fundamentally, the methods pose different questions; % cover provides a picture of the overall coral community composition and enables comparison across other competing benthic taxa (algae, sponges, etc.), whereas demographic surveys are necessarily species inventories aimed at characterizing populations. For sampling of deep water reefs, the decision to use photoquads, demography transects, or both methods should be based on the underlying questions motivating the survey.

Demographic and percent cover approaches can both pose pitfalls. For example, increasing frequencies of small colonies can result from beneficial processes such as recruitment of new colonies or undesirable processes such as partial mortality resulting in small remnant colonies. Meanwhile, a single massive colony is not functionally equivalent (in terms of habitat value, susceptibility to various threats, nor fecundity) to many small colonies but might be represented as equal in terms of percent cover. Photoquadrats do not necessarily sample every coral in the area surveyed, and can have poorer resolution or detectability for rare, small, or similar-appearing species. For example, we were not able to quantify *L. cucullata* abundance reliably from our photos (likely mis-identified as *Agaricia* spp., or present in cryptic locations not visible in top–down photographs) whereas it was easily distinguishable in the field. Hence, the significant loss of this species (sixth most common taxon in 2004) would not have been detected from photoquadrat sampling alone. The decline of *L. culcullata*, also reported in Curaçao (*Vermeij et al., 2011*) and Jamaica (*Hughes & Tanner, 2000*), is likely the most substantial collapse of a Caribbean coral species since *Acropora* spp. but has largely gone unnoticed due to predominance of photographic monitoring approaches.

Most long-term coral monitoring efforts have relied solely on percent cover to quantify abundance, community structure, and changes through time. This approach has been sufficient to detect changes over long time frames (*De'ath et al., 2012*) and the drastic losses over short time frames due to recent acute disturbances on Atlantic/Caribbean reefs (e.g., *Coelho & Manfrino, 2007*; *Miller et al., 2009*). Significant declines are relatively easy to detect from a baseline of 50% cover, but change detection likely requires much greater sampling effort from a baseline signal of 10%, or even much lower for individual coral species, as characterizes most modern Caribbean reefs (e.g., *Gardner et al., 2003*; *Ruzicka et al., 2013*). Within the photoquadrat data set reported here, significant change is detectable over less than a decade in coral percent cover and even in the few dominant individual coral taxa (Fig. 2). However, given the low percent coral cover in Navassa by the end of the study period, it seems highly unlikely that future changes in total coral cover (either continued decline from a low baseline or, hopefully, recovery), let alone individual taxa, will be detectable over the next decade without substantially greater sampling effort if relying on photoquadrat sampling alone. For example, *Molloy et al. (2013)* performed intensive power analyses to determine the number of photoquadrats/points required to

detect 1% per annum recovery in coral cover and concluded this scale of recovery was essentially impossible to detect with their most intensive photoquadrat protocols (250 quadrats per site, 50 points per quadrat). While traditional percent cover data such as from photoquadrats provides crucial information on the status of the 'other 90%' of the reef that is not hard coral, it may provide relatively less information for corals, especially when they are at low abundance. By supplementing photoquadrat data with minimal demographic sampling we were able to detect increased densities of several species over eight years. The collection of colony-based (i.e., demographic) data provides additional metrics, greater resolution and analytical power (e.g., hundreds of colonies for many taxa in an effort such as this, rather than a cover estimate of, e.g., <1%), and a valuable mechanistic insight as to the population dynamics driving coral population changes. Hence, a combined approach employing both photoquadrat data with demographic data may be the most informative to track changes in benthic reef communities at low coral abundances.

## ACKNOWLEDGEMENTS

This work was made possible by the excellent logistic support provided by D McClellan, J Javech, the John G. Shedd Aquarium's R/V Coral Reef II, the NOAA Ship Nancy Foster, and the LOF's M/Y Golden Shadow. Special thanks to A Bruckner (LOF) for enabling 2012 data collection.

### Funding

This work was funded by the NOAA Coral Reef Conservation Program and in kind support from the Southeast Fisheries Science Center, the John G. Shedd Aquarium and the Khalid bin Sultan Living Oceans Foundation. The funders had no role in study design, data collection and analysis, decision to publish, or preparation of the manuscript.

### Grant Disclosures

The following grant information was disclosed by the authors:
NOAA Coral Reef Conservation Program.
Southeast Fisheries Science Center.
Khalid bin Sultan Living Oceans Foundation.

### Competing Interests

The authors declare there are no competing interests.

### Author Contributions

- Margaret Miller and Dana E. Williams conceived and designed the experiments, performed the experiments, analyzed the data, wrote the paper, prepared figures and/or tables, reviewed drafts of the paper.
- Brittany E. Huntington and Gregory A. Piniak conceived and designed the experiments, performed the experiments, wrote the paper, reviewed drafts of the paper.

- Mark J.A. Vermeij conceived and designed the experiments, performed the experiments, contributed reagents/materials/analysis tools, wrote the paper, reviewed drafts of the paper.

**Field Study Permissions**

The following information was supplied relating to field study approvals (i.e., approving body and any reference numbers):

US Fish and Wildlife Service, Navassa National Wildlife Refuge Special Use Permits # 41529-2002-10, #41529-2004-12 and #41529-2006-03, #41529-2009-01, and #41529-2012-001.

**Data Availability**

Two files of raw data have been uploaded as Supplemental Information.

**Supplemental Information**

Supplemental information for this article can be found online at http://dx.doi.org/10.7717/peerj.1643#supplemental-information.

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
