# Peer review of "Decadal comparison of a diminishing coral community: a study using demographics to advance inferences of community status"

_PeerJ, doi:10.7717/peerj.1643_

## Round 0.1 · original submission · Major Revisions

Please go through the comments made by each reviewer. When you submit your revised ms, please also provide detailed response (prefer to be point to point) to their comments. Your revised manuscript wil be subject to further review.

Reviewer 1 ·

Basic reporting

The paper presented by Miller et al address an important topic. The authors point on the importance of demographic data highlighting the higher amount of information that can be obtained by this kind of sampling, compared to the classical estimation of % cover. I agree with the authors on this point, also if they do not mention enough the fact that photo sampling for % cover allow way more quantity of data than demographic sampling. Authors claim the same amount of underwater work is needed but it seems unrealistic, given that for a demographic sampling, each colony have to be measured on site, while the photographic sampling for % cover does not require nothing more that shooting pictures. This is particularly valid in "deep" environment (30 m) in which time constraints due to SCUBA diving limits does not allow too many in situ measurements.

Experimental design

Regarding the experimental design, a problem that I found in this manuscript is the fact that the experimental design is too much "arranged". Demographic data, for example, have bee taken each year with a different methods and a lot of adjustments have been done in order to make the data comparable, adding uncertainty to the results.

Another issue comes form the different timing of data: % cover refers to the period 2002 - 2012 while demographic data refers to the period 2004-2012. In order to compare the two methods, authors should have at least taken into account the same time frame, analysing only % cover data in 2004 and 2012. Moreover authors state that a big decrease in % cover happened between 2002 and 2004, just in the period that is not comparable with demographic data. Why not analysing the same time frame for the two methods?

Regarding the statistics applied, I have a concern about the use of ANOVA by ranks (in the case of % cover) and Mann-Whitney rank sum test (in the case of demographic data) for temporal series. When analysing temporal series, data are not independent between the subsequent years and there are statistical methods to overcame this problems i.e. repeated measures ANOVA or Friedman test.

Validity of the findings

As I told in the introductory comment, I agree with the idea of the manuscript and I am convinced that demographic data can give more and more precise information, but I doubt that the data the authors present can clearly demonstrate this, for the reasons I stated in the comments regarding the experimental design.

Another not clear point is related to the changes in population structure. Authors found a significant loss of coral cover but a coral density that remained unchanged. If cover decrease and density remains unchanged, it means that size is decreasing. In the results it is not clear if size is increasing or decreasing. In line 228 authors state that some species shifted towards larger colonie and some other species shifted towards smaller size. Then they comment (lines 231 and following) that the most dramatic (I should use the word drastic, more than dramatic) difference is size distribution is a shift to larger colonies in Porites. All this part is not extremely clear and authors should try to explain why cover decrease why density does not change, which is, in my opinion, an interesting outcome that deserves to be more deeply considered. Is it possible, for example, that this result derive from the fact that % cover is analysed between 2002 and 2004 when a big decrease happens, while density refers only to 2004 and 2012?

Or could be possible that the use of two different methods to size the colonies between 2004 and 2012 could result in an apparent stability of density?

All these points should be clarified before the publication of the manuscript

Additional comments

As I told already, I support the idea of the paper, I completely agree on the utility of demographic data, but I think that the data presented in this manuscript, in the form they are presented, cannot support the ideas proposed by the authors. Notwithstanding i believe that with revision of some statistical approach and with a deeper analysis of the data, some conclusions could be reached, for this reason I propose a mayor revision.

·

Basic reporting

No comment – based on the comment by the authors that the raw data will be uploaded upon acceptance – which I believe is appropriate.

Experimental design

In the abstract and Intro and methods, the sites are described as being selected opportunistically and haphazardly and randomly. I am not sure if there is a difference. I suggest that a bit more information – if possible- on how sites were selected would be of interest to the readers – this is especially true in reference to my next comment.

The fact that the photo sites and demo sites were not the same is important when interpreting the results. I understand the constraints associated with sampling remote areas and the difficulty in sampling the ‘same sites’ over multiple years. I did not really understand that the sites were not the same until line 100. I think it would help the reader to make this clearer in the last paragraph of the Intro.

Generally how far apart where the 8-10 photoquads at each site?

For large colonies – especially Orbicella colonies – defining a ‘colony’ can be challenging. Some programs use actual colony boundaries to estimate ‘colonies’ while others – such as yours – use isolates as ‘colonies’. It was good that you clearly stated this approach, but why did you choose this approach?

There does not appear to be a figure legend for Figure 4.

In Suppl Image 1 there does not appear to be as many sites shown as number of sites indicated in Table 1. From the table it looks like there were 50 sites sampled (2004 + 2012) along the SW coast but there does not seem to be 50 symbols. Is it possible to differentiate the photo sites from the demo sites on the map?

Validity of the findings

No comments.

Additional comments

This is a very well written manuscript that should be accepted. The minor suggested edits/comments above were basically made to help make some points a bit more clear. This type of comparison should be completed by other programs (including my own) which have the luxury of having both photo derived cover data and demographic data from the same sites over multiple years.

Reviewer 3 ·

Basic reporting

The subject matter for this article is worthy of publication because there is little information available on reef condition on this uninhabited island in the Caribbean. The opportunity for comparing reef condition with other areas that experience land-based pollution is certainly worth exploration. However, the article instead focuses on a comparison of methods and trend detection. The many caveats in sampling design, sampling locations, and methodology, only partially annotated in the first paragraph of the discussion section, preclude defensible interpretations for either topic. Comparison of methods, other than to highlight the different endpoints, requires that at least some stations are assessed by both methods at the same time. Similarly, trend detection requires at least a few repeated locations (to avoid confusion of temporal with spatial variability) and a consistent methodology. This is particularly important for coral reefs which have a highly patchy distribution.
For these major reasons, the manuscript is unacceptable in its present form. However, with a slightly different approach, the disparate data sets could prove useful. If the data were presented simply as observations (not method comparison or trend detection), they could be generically compared to reef condition in other parts of the Caribbean to shed light on effects of land-based anthropogenic stressors. There are numerous publications using live coral cover in the Caribbean and both NOAA and EPA have completed at least a few demographic surveys in the Caribbean (see references below), although these studies use diameter and height measurements rather than only diameter.

Experimental design

See above; this report incorporated several different experimental designs and methods which contributed to lack of defensibility.

Validity of the findings

See above; the various experimental designs, methods, and treatment of data made the inferences and interpretations largely unacceptable. A major factor is no control for spatial heterogeneity in a community with historic patchiness.

Additional comments

The authors could consider the suggestion above re: comparison with existing Caribbean reef condition assessments. In addition, there were also many minor concerns, some noted below;
The abstract indicates (line 22) that demographic measures would “provide better resolution and mechanistic inference of processes shaping coral populations through time”. This is not revisited in the discussion. However, shifts in colony size explained by fission or recruitment seems to be an inference; is there more? Also, it is never made clear how demographic sampling would be valuable in logistically constrained sampling efforts (line 23).
Based on reef heterogeneity, the shift from haphazard to random sampling (line 90) generates concern over any comparison among the data—haphazard could mean that divers went to where actual reefs occurred and random means they may have sampled areas that were not necessarily ‘reef’. The differences could be very important in the decline in condition reported and interpreted.
Wasn’t clear how methods for photoquadrat data were applied for the three most abundant species (line 93). Were additional point counts made for these species? And for the three rarer species (line 97)?
Adjusting two-dimensional area for a colony with three-dimensional %live surface area estimates seems problematic (line112).
If north coast reefs are substantively distinct from the southwest reefs (line 122), how defensible is it to include them in analyzing changes in size structure (line 139) just to increase n?
It is difficult to understand how such an important species and abundant (Porites astreoides) could be excluded from size frequency analyses (line 145). This undermines the community structure emphasis.
How can partial mortality cause the loss of Orbicella colonies (line 229)? Loss of live coral cover, yes, but the colony?
Throughout the results and discussion, the words increase, decrease, change and shift imply that there was continuity in sampling design and locations; it may be more appropriate to use higher, lower and different since there was no such continuity. Much of the discussion is spent trying to explain ‘changes’ observed over time, when in fact the results may only reflect spatial variation.
Stephanocoenia intersepta (not intercepta) in discussion; could also spell out L. cucullata first time outside of abstract.

---

## Round 0.2 · Minor Revisions

The authors need to address or at least discuss the shortcomes of the current sampling methodology in the revised version.

Reviewer 1 ·

Basic reporting

I see that the manuscript was modified trying to address the suggestion of the referees and it is actually better than in the first version. Regarding the different effort and efficiency of photographic vs in situ sampling I agree that the pictures method require a considerable amount of time of computer analysis, on the other side there’s the deep sampling for which the in situ sampling is less efficient due to the short bottom times, and this should be mentioned in the manuscript.

I noticed that the authors tried to specify that the paper is not intended to be a comparison between the two methods of sampling, which was one of the not very well explained parts of the previous version, but I think that this idea is still not out of the manuscript. Reading the introduction, for example, it looks like that the objective is a comparison between the two methods and only in line 67-72 the authors says that they want to analyze the data together in order to answer the questions. I suggest to make the rest of the introduction more fitting with the objectives declared in lines 67-72. The same is valid for the last paragraph of discussion (lines 341-348) where again it looks like the author aim at comparing the two methods, which I think it is not the objective of the paper.

Finally the authors changed the manuscript following the directions of the reviewer and I think that, with some small changes, the manuscript can be published.

Experimental design

Experimental design still have the flaws of being arranged, but authors have better made it clear it in the manuscript.
Regarding the statistical analysis I recognize that ANOVA by ranks is the correct analysis for temporal data and my comment was due to a misinterpretation.

Validity of the findings

Validity of the findings are also affected by the nature of a not planned sampling design, but this is also now better stated in the manuscript. Probably a clear statement on the caveats related to not homogeneous and not planned sampling design could help the reader.

Additional comments

As I told in the previous revision of this manuscript, I support the idea that demographic data are extremely useful to have a better understanding of the dynamics of populations.

---

## Round 0.3 · accepted · Accept

Thank you for your effort in further revising your ms. I am happy with the changes made to address the issues raised by the review and glad to accept this ms for publication.